# Spatial patterns and clustering of dengue incidence in Mexico: Analysis of Moran's index across 2,471 municipalities from 2022 to 2024

Oliver Mendoza-Cano[1], Rogelio Danis-Lozano[2], Xóchitl Trujillo[3], Miguel Huerta[3], Mónica Ríos-Silva[4], Agustin Lugo-Radillo[5], Jaime Alberto Bricio-Barrios[4], Verónica Benites-Godínez[6,7], Herguin Benjamin Cuevas-Arellano[8], Juan Manuel Uribe-Ramos[1], Ramón Solano-Barajas[1], Yolitzy Cárdenas[3], Jesús Venegas-Ramírez[9], Eder Fernando Ríos-Bracamontes[10], Luis A. García-Solórzano[11], Arlette A. Camacho-delaCruz[1], Efrén Murillo-Zamora[4,12]*

1 Facultad de Ingeniería Civil, Universidad de Colima, Coquimatlán, Colima, México, 2 Centro Regional de Investigación en Salud Pública, Instituto Nacional de Salud Pública, Tapachula de Córdova y Ordóñez, Chiapas, México, 3 Centro Universitario de Investigaciones Biomédicas, Universidad de Colima, Colima, Colima, México, 4 Facultad de Medicina, Universidad de Colima, Colima, Colima, México, 5 SECIHTI—Facultad de Medicina y Cirugía, Universidad Autónoma Benito Juárez de Oaxaca, Oaxaca, México, 6 Coordinación de Educación en Salud, Jefatura de Servicios de Prestaciones Médicas, Instituto Mexicano del Seguro Social, Tepic, Nayarit, México, 7 Unidad Académica de Medicina, Universidad Autónoma de Nayarit, Tepic, Nayarit, México, 8 Facultad de Ciencias, Universidad de Colima, Colima, Colima, México, 9 Coordinación Auxiliar Médica de Investigación en Salud, Jefatura de Servicios de Prestaciones Médicas, Instituto Mexicano del Seguro Social, Colima, Colima, México, 10 Departamento de Medicina Interna, Hospital General de Zona No. 1, Instituto Mexicano del Seguro Social, Villa de Álvarez, Colima, México, 11 Tecnológico Nacional de México, Campus Colima, Villa de Álvarez, Colima, México, 12 Unidad de Investigación en Epidemiología Clínica, Instituto Mexicano del Seguro Social, Villa de Álvarez, Colima, México

* efren.murilloza@imss.gob.mx

## Abstract

Dengue is an increasing public health challenge, with rising cases and expanding distribution. Its complex epidemiology is influenced by climate change, urbanization, and the circulation of multiple viral serotypes. This study aimed to characterize the spatial and temporal (2022–2024) patterns of dengue incidence across 2,471 municipalities in Mexico. Weekly case counts, obtained through the normative epidemiological surveillance system for vector-borne diseases, were used to calculate incidence rates per 100,000 population. Geographic Information Systems were employed to analyze spatial patterns, while Local Moran's I statistic and a k-nearest neighbors spatial weights matrix identified spatial clusters. A total of 622,689 dengue cases were analyzed, with incidence rates rising from 29.4 in 2022 to 279.0 per 100,000 in 2024. Dengue transmission expanded, affecting 38.0% of municipalities in 2022 and 68.6% by 2024 ($p < 0.001$). Spatial clustering also increased, with positive clusters increasing from 28 municipalities in 2022–98 in both 2023 and 2024. Moran's I values indicated a peak in spatial autocorrelation in 2023 ($I = 0.57$). While DENV-2 was the predominant serotype in 2022, DENV-3 became dominant in 2023 and 2024. Over

**Data availability statement:** The original data presented in the study are openly available in the following repository: https://www.gob.mx/salud/documentos/datos-abiertos-bases-historicas-direccion-general-de-epidemiologia

**Funding:** The author(s) received no specific funding for this work.

**Competing interests:** The authors have declared that no competing interests exist.

time, high-incidence areas shifted from southern and central regions to the southeast and Pacific coast. These findings suggest the growing burden of dengue in Mexico, driven by rising incidence, expanding geographic distribution, and evolving spatial patterns. A coordinated public health response is needed to mitigate the impact of dengue and prevent its spread to newly affected areas.

## Introduction

Dengue fever, a vector-borne viral disease primarily transmitted by *Aedes aegypti* mosquitoes, is a major public health challenge globally, particularly in tropical and subtropical regions. The World Health Organization has recognized dengue as the most prevalent vector-borne viral disease, with an estimated 50–400 million infections occurring annually [1]. The incidence of dengue is influenced by various factors requiring a comprehensive understanding of its spatial and temporal patterns to inform effective public health interventions [2].

Dengue transmission is often concentrated in specific geographic areas, known as hotspots or persistent transmission zones, shaped by human behavior and environmental conditions (ecological, entomological, infrastructural, and social) [3]. Climatic factors, such as temperature and rainfall, affect mosquito breeding, survival, and viral replication [4]. Higher temperatures accelerate mosquito development, shorten the extrinsic incubation period of the virus, and enhance vector activity [5]. Meanwhile, rainfall creates breeding sites by accumulating water in natural and artificial containers, increasing mosquito populations [6].

Vector biology plays a significant role in the transmission of dengue virus (DENV). The most well-known mechanism is horizontal transmission, where the virus is passed alternately between humans and mosquitoes. However, vertical transmission, which occurs when infected female mosquitoes pass the virus to their offspring, may help explain the persistence of DENV in nature during periods without infected humans or under unfavorable environmental conditions for adult mosquito survival [7].

Dengue serotypes shape spatial and temporal patterns by influencing immunity, transmission, and outbreak severity [8]. Serotype shifts can drive new epidemics, alter geographic spread, and impact disease severity through immune interactions [9]. The spatial distribution of dengue cases has also been linked to urbanization and population density. Urban areas exhibit higher incidence rates due to increased interactions between humans and mosquitoes, creating diverse and heterogeneous ecological transmission niches [10,11]. Temporal dynamics are crucial in the clustering of dengue cases.

Irregular seasonal patterns (interepidemic) in dengue incidence complicate the application of traditional time series models, highlighting the value of spatial analyses for understanding disease risk and transmission dynamics [12]. Studies employing advanced statistical methods, such as spatial autocorrelation and cluster analysis, underscore the interplay between spatial and temporal factors in dengue outbreaks [13,14]. For example, the use of Moran's I statistic has revealed significant spatial

autocorrelation in dengue incidence, demonstrating that cases are concentrated in specific areas rather than randomly distributed [15,16].

This research aimed to characterize spatial and temporal patterns of dengue incidence across 2,471 municipalities, identify high-incidence areas, and evaluate trends that could inform public health interventions and resource allocation for dengue prevention and control at the local level. We hypothesize that dengue transmission hotspots are not randomly distributed but follow spatial patterns influenced by disease dynamics over time. Specifically, we expect that municipalities with historically high dengue incidence will exhibit persistent transmission, with spatial clustering reflecting underlying transmission trends.

## Materials and methods

### Study design and data source

This study analyzed open data from the General Directorate of Epidemiology in Mexico to examine dengue incidence from 2022 to 2024. The dataset included weekly dengue case counts from 2,471 municipalities across Mexico's 32 states, obtained through the normative epidemiological surveillance system for vector-borne diseases [17]. Cases were classified as suspected or confirmed according to the clinical and laboratory criteria established by the Directorate of Epidemiology. Confirmed cases were identified through reverse transcription polymerase chain reaction (RT-PCR) testing of venous blood samples or serological assays (for patients with five or more days between symptom onset and blood sampling).

### Data preparation and rate calculation

The dengue incidence rates per municipality were calculated by dividing the number of reported cases by the population of each municipality, multiplied by 100,000 inhabitants. Population data were obtained from publicly available projections by the National Population Council of Mexico [18]. No imputations were performed, and municipalities with missing incidence data were excluded. To facilitate spatial comparisons, municipalities were categorized into quintiles based on the computed rates.

### Spatial analysis and mapping

Mapping was performed in R 4.4.1 (R Core Team) using the ggplot2 package [19]. The shapefile used as the base map was developed by the research team using publicly accessible geographic coordinates of municipal boundaries obtained from the Geostatistical Framework 2024, provided by the National Institute of Statistics and Geography (INEGI) of the Mexican government [20].

Each municipality's dengue incidence rate was linked to the corresponding geographic polygon in the shapefile, allowing spatial visualization of high-incidence areas. Choropleth maps were generated to display dengue incidence rates, highlighting municipalities with varying levels of incidence.

### Statistical analysis

Summary statistics were computed. Spatial autocorrelation analysis was conducted using Local Moran's I statistic to identify clusters of municipalities with similar incidence rates and detect spatial outliers. This index quantifies the degree of spatial dependence at the local level, allowing for the identification of positive (high-high or hotspots) or negative (low-low or cold spots) autocorrelations clusters.

Local Moran's I was calculated using a spatial weights matrix, with the number of k-nearest neighbors (kNN) fixed at 5 to define spatial relationships. This choice was justified by considering Mexico's geographic and epidemiological diversity, ensuring that each region is compared with a consistent number of neighboring areas despite variations in population density and transmission patterns [21]. This approach enhances the reliability of identifying spatial clusters across diverse

settings [22]. Statistical significance was evaluated through permutation tests with 999 iterations, and p-values < 0.05 were considered significant. All statistical analyses were conducted using R 4.4.1 (R Core Team) and the sfdep package [23].

### Ethics statement

This study utilized publicly available, de-identified data; therefore, no ethical approval was required. All data processing and analyses complied with applicable data protection regulations.

## Results

A total of 622,689 dengue cases were analyzed. The incidence rates in 2022, 2023, and 2024 were 29.4, 170.6, and 279.0 cases per 100,000 population, respectively. In the analyzed period, dengue transmission was reported in 940 (38.0%), 1,439 (58.2%), and 1,696 (68.6%) municipalities, respectively, reflecting a statistically significant increase over time ($p < 0.001$; chi-square statistic = 483.2). Fig 1 presents the week of symptom onset of the analyzed suspected ($n = 462,804$) and PCR-confirmed dengue cases ($n = 159,885$).

### Yearly dengue incidence rates

Fig 2a-c presents dengue incidence rates across the 2,471 analyzed municipalities in Mexico from 2022 to 2024. The calculated incidence rates for each municipality and year are detailed in S1 Dengue incidence rates by municipality, 2022–2024.

The highest dengue incidence rates in 2022 (Fig 2a) were mainly concentrated in municipalities within Western, Southern, Northwestern, and Central Mexico. The municipalities with the highest incidence rates (per-100,000) included Ónavas, Sonora (3,835.6), Álamos, Sonora (3,451.3), Tejupilco, Estado de México (2,180.8), Navojoa, Sonora (2,050.6), Luvianos, Estado de México (1,835.4), San Bartolo Yautepec, Oaxaca (1,378.3), San Miguel de Horcasitas, Sonora (1,370.1), Huatabampo, Sonora (965.5), and Etchojoa, Sonora (942.8). Serotype data were available for 6,210 patients, revealing the following distribution: DENV-2 (57.0%), DENV-3 (23.9%), DENV-1 (17.8%), and DENV-4 (1.3%).

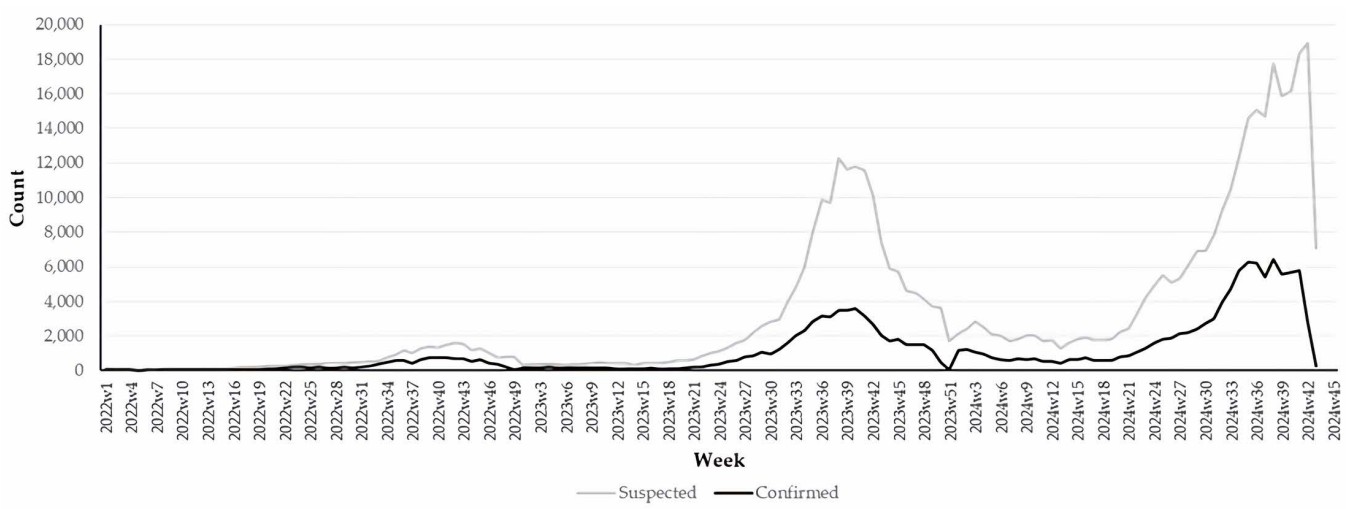

**Fig 1. Weekly distribution of analyzed dengue cases by classification status, Mexico 2022–2024.** Notes: 1) Suspected cases ($n = 462,804$) met clinical criteria but did not undergo analytical procedures per normative guidelines, while confirmed cases ($n = 159,885$) had a positive result from reverse transcription polymerase chain reaction; 2) Data for 2024 includes cases up to week 43 (October 26, 2024).

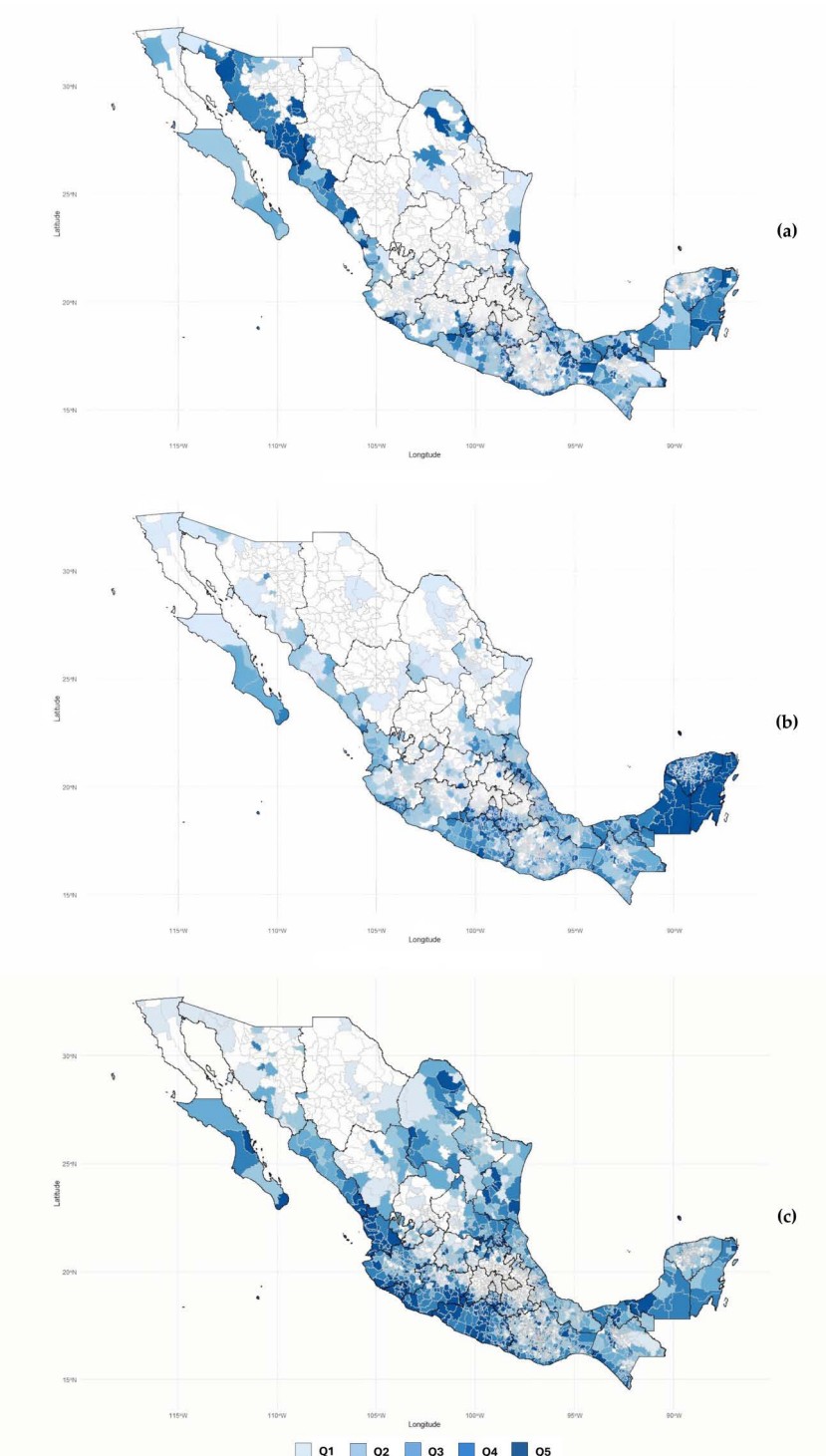

**Fig 2. a-c. Dengue incidence rates (per 100,000 population) across 2,471 municipalities in for the years 2022 (a), 2023 (b), and 2024 (c), Mexico.** Notes: 1) Quintile ranges for dengue incidence rates (per 100,000 population) were as follows: 2022, Q1 (0.1–6.8), Q2 (6.9–19.5), Q3 (19.6–44.8), Q4 (44.9–103.7), Q5 (103.8–3,835.6); 2023, Q1 (0.1–14.5), Q2 (14.6–60.9), Q3 (61.0–184.3), Q4 (184.4–515.0), Q5 (515.1–6,192.2); 2024, Q1 (0.2–28.5), Q2 (28.6–87.9), Q3 (88.0–210.8), Q4 (210.9–539.5), Q5 (540.0–6,677.5); 2) Municipalities without registered dengue cases are shown in white.

In 2023, as shown in Fig 2b, the highest dengue rates were observed in southeastern Mexico, an area characterized by its proximity to the Gulf of Mexico and the Caribbean Sea and as part of the Yucatán Peninsula. Elevated rates were also recorded in Puente de Ixtla, Morelos (6,192.2), a municipality in central Mexico, as well as in areas along the South Pacific, specifically in San Pedro Juchatengo, Oaxaca (4,900.3), and Buenavista de Cuéllar, Guerrero (4,552.5). During 2023, DENV-3 replaced DENV-2 as the most frequently identified viral serotype compared to the 2022 scenario. DENV-3 was documented in 12,668 (59.0%) of 21,467 patients with an identified pathogen. The remaining frequencies were 22.0% for DENV-2, 16.9% for DENV-1, and 2.2% for DENV-4.

High dengue rates were observed along the Pacific coast during 2024, particularly in the Western, Southern, and Southeastern regions (Fig 2c). Similar patterns were also found in the Northwest region and the coastal areas of the Gulf of Mexico. The highest incidence rates, in descendent order, were documented in Acteopan, Puebla (6,677.5), Minatitlán, Colima (6,265.3), Atzala, Puebla (4,894.2), Peñamiller, Querétaro (4,691.5), and Armería, Colima (4,626.1).

Also in 2024, data regarding the infectious serotype was available for 42,545 patients. DENV-3 was the most frequently identified (86.2%), followed by DENV-1 (7.6%), DENV-2 (4.8%), and DENV-4 (1.4%).

### Spatial clustering of dengue incidence rates

Key municipalities with significant clusters (2022–2024) are listed in S2 Clustering data, along with their Moran's I statistic, Z-scores, and geographic coordinates. The total number of municipalities with significant spatial autocorrelations was 49 in 2022, 113 in 2023, and 119 in 2024, corresponding to 5.2%, 7.9%, and 7.0% of all municipalities reporting dengue cases in each respective year.

The overall Moran's I values were 0.27 (s²=0.0001) in 2022, 0.57 (s²=0.0002) in 2023, and 0.49 (s²=0.0002) in 2024, indicating an increase in spatial clustering of dengue incidence in 2023. The magnitude of Moran's I reflects the degree of clustering, where values closer to +1 indicate strong clustering and values near -1 indicate strong dispersion. The *p*-value from the Moran's I was 2.2e-16 in all the three analyzed years.

Positive clusters were observed in 28 municipalities in 2022, increasing to 98 municipalities in both 2023 and 2024. The number of negative and significant clusters identified annually were 21 in 2022, 15 in 2023, and 21 in 2024. Fig 3a–c maps the spatial distribution of these significant clusters.

As shown in Fig 3a, positive clusters during 2022 were predominantly located in southern and central Mexico, specifically in Guerrero and Puebla, two neighboring states. These clusters alternated with significant negative clusters within the same areas, highlighting a pattern of spatial heterogeneity. Additional positive clusters were observed in another central state, the State of Mexico, particularly in its southern region bordering Guerrero and Michoacán. This area also exhibited an alternation between positive and negative spatial autocorrelations. Other significant positive clusters were documented in regions of the northwest and south of the country.

In 2023 (Fig 3b), significant positive clusters were identified in central Mexico, particularly in Morelos and Puebla, as well as in the southern state of Guerrero, which also exhibited notable cold spots. Additional positive clusters were observed in the southeast, concentrated in the Yucatán Peninsula. Meanwhile, cold spots were predominantly found along the coastal region of Oaxaca in southern Mexico and in parts of the central region of the country.

Finally, in 2024, positive clusters were documented in west-central Mexico, particularly in the states of Colima, southern Jalisco, and Nayarit. Other significant positive clusters were identified in the central region, in Morelos, as well as in the southern states of Guerrero and Oaxaca. Negative and significant clusters were observed interspersed within many of these positive clusters.

## Discussion

Our study analyzed dengue incidence patterns in 2,471 municipalities in Mexico over three consecutive years (2022–2024), revealing an increase in registered cases and geographic spread. The observed rise in national dengue incidence

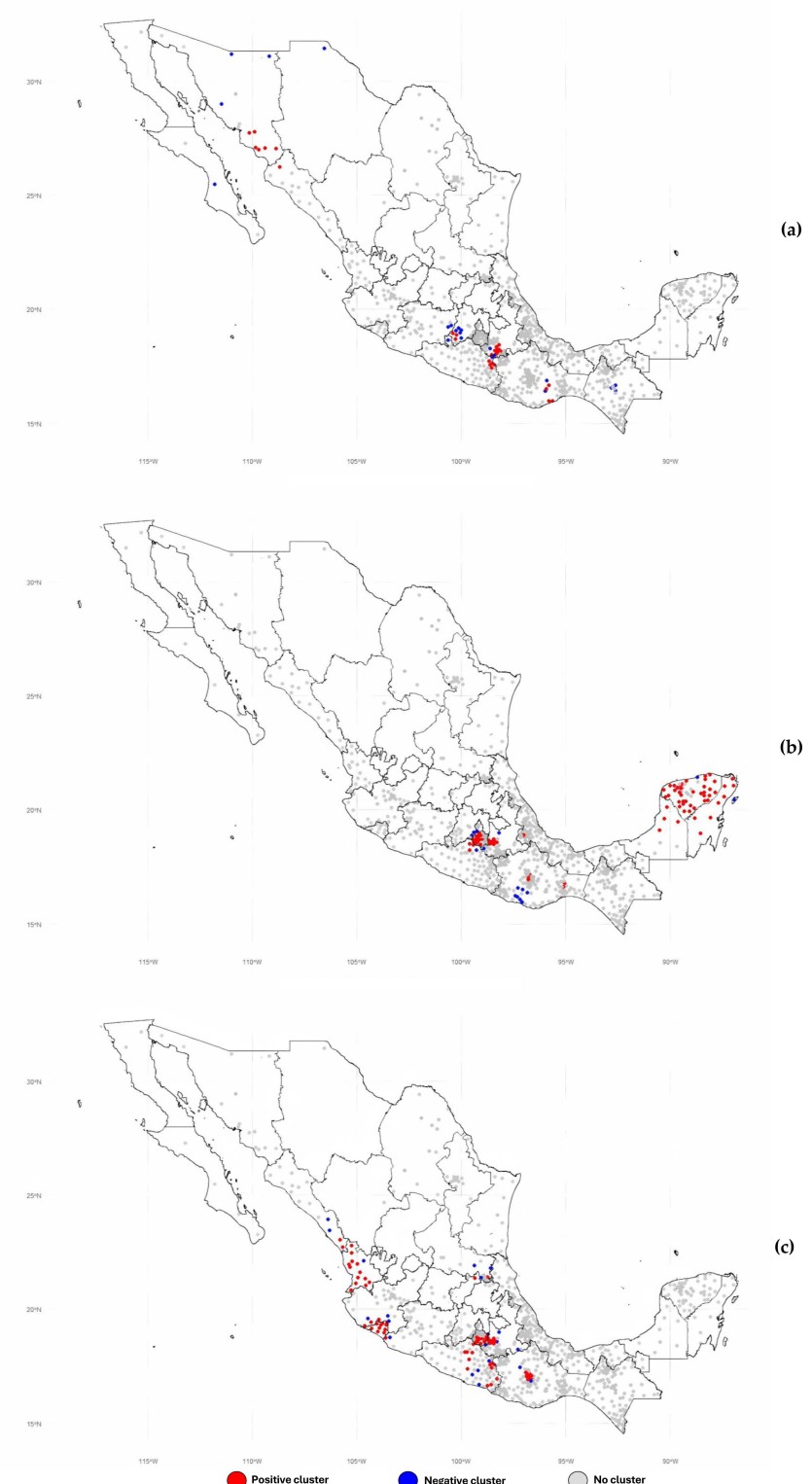

**Fig 3. a-c. Dengue incidence rates (per 100,000 population) across 2,471 municipalities in for the years 2022 (a), 2023 (b), and 2024 (c), Mexico.**

rates, from 29.4 cases per 100,000 population in 2022 to 279.0 per 100,000 in 2024, underscores the escalating public health burden posed by dengue.

Another key finding is the significant increase in the number of municipalities reporting cases over the study period, which may indicate an expansion of dengue transmission zones and highlights the growing geographical heterogeneity of the disease [24]. For instance, Lee et al. demonstrated that the expansion of dengue transmission zones in Brazil is significantly associated with temperature suitability, urban connectivity, and urbanization processes, indicating that these factors enhance the likelihood of future outbreaks once both the vector and the virus are introduced into new areas [25]. The impact of climate change on dengue transmission has been documented. Published data suggests that climate warming could significantly alter the geographic and seasonal ranges of dengue, with the potential for increased transmission in previously unaffected areas [26].

Spatial analyses indicated clustering of dengue incidence, with significant positive clusters increasing from 28 municipalities in 2022–98 municipalities in both 2023 and 2024. The Moran's I statistics evidenced an increase in spatial autocorrelation from 2022 (I = 0.27) to 2023 (I = 0.57), followed by a slight decrease in 2024 (I = 0.49). These findings suggest that while dengue cases became more geographically concentrated in 2023, the clustering pattern evolved in 2024, likely influenced by local transmission dynamics, ecological factors, and human behavior [27,28]. The findings support the hypothesis that dengue transmission hotspots are not randomly distributed but follow spatial and temporal patterns. The shift in high-incidence areas from the southern and central regions to the southeast and Pacific coast highlights the evolving nature of dengue distribution.

Regionally, the spatial distribution of positive clusters revealed distinct trends. In 2022, clusters were primarily concentrated in southern and central Mexico, particularly in Guerrero, Puebla, and the State of Mexico. In 2023, clustering shifted to include southeastern regions, particularly the Yucatán Peninsula, and central states like Morelos. The geographic redistribution of dengue clusters coincided with a shift in dominant serotype from DENV-2 in 2022 to DENV-3 in 2023, suggesting that viral dynamics may have contributed to the observed spatial patterns [29,30]. By 2024, positive clusters were most prominent along the Pacific coast, in west-central states like Colima and Jalisco, and in central and southern regions, including Guerrero and Oaxaca, further underscoring the evolving landscape of dengue transmission in Mexico. The dominance of DENV-3 in 2023 and 2024, replacing DENV-2, suggests that serotype shifts may contribute to these spatial and temporal trends.

In Mexico, over the past decade, dengue cases concentrated in regions characterized by tropical climates, high humidity, and abundant rainfall, creating ideal conditions for the proliferation of *Ae. aegypti* [31]. Recent data indicate a gradual northward expansion of dengue cases in the country [32].

Southeast Asia, particularly countries like Thailand, Indonesia, and the Philippines, has historically been a hotspot for dengue [33]. Like Mexico, these regions experience high dengue incidence due to tropical climates and dense urban populations [34].

In South America, Brazil stands out as the country with the highest dengue burden [35]. The spatial distribution of dengue in Brazil mirrors that of Mexico, with cases concentrated in urban and peri-urban areas [36]. Unlike Mexico, Brazil has also reported the emergence of dengue serotype shifts, complicating control efforts [37].

In Sub-Saharan Africa is underreported but is increasingly recognized as a growing problem. Countries in Sub-Saharan Africa, like Nigeria and Kenya have reported sporadic outbreaks, often in urban centers [38]. The spatial trends in this region differ from Mexico, as dengue transmission is less predictable and often overlaps with other mosquito-borne diseases like malaria [39].

The Caribbean and Central American countries, such as Puerto Rico and Honduras, exhibit dengue trends like Mexico, with coastal and low-lying areas being most affected. These regions share comparable environmental and climatic conditions, but the Caribbean has experienced more frequent outbreaks and introduction of new dengue serotypes due to high tourist mobility [40].

The observed shift in high-incidence areas of vector-borne diseases raises questions about the underlying drivers of this epidemiological transition. This is likely driven by a complex interplay of climate variability, vector control measures, and urbanization patterns [41]. Addressing this shift requires a multifaceted approach that includes strengthening vector control infrastructure in emerging high-risk regions, enhancing surveillance systems to monitor climatic and epidemiological trends, and promoting sustainable urban planning to reduce vector habitats [42].

The serotype distribution further highlighted significant shifts in viral epidemiology over time. While DENV-2 predominated in 2022, DENV-3 emerged as the dominant serotype in 2023 and 2024, with an increasing prevalence. This shift may be linked to changes in population-level immunity, as the circulation of DENV-3 had not been documented in Mexico for over a decade [43]. Other potential determinants include viral evolution, differences in mosquito populations, and environmental conditions. Since each serotype can impact disease severity and outbreak patterns differently, understanding these potential determinants is key to developing effective public health strategies [44–48].

Serotype shifts in DENV circulation have public health implications, as they may influence disease severity and vaccine effectiveness. The emergence of a previously less dominant serotype can lead to an increased burden of severe disease, particularly in populations with prior heterotypic immunity. Additionally, such shifts may affect the performance of existing vaccines, which rely on balanced protection against all four serotypes. Continuous epidemiological and immunological surveillance is essential to anticipate these challenges and inform adaptive vaccination strategies.

From a public health perspective, understanding the temporal and geographical trends of dengue in Mexico from 2022 to 2024 is crucial for optimizing disease prevention and control strategies. The spatial distribution of dengue positive clusters underscores the influence of environmental and sociodemographic factors on disease dynamics. Temporal trends, marked by recurrent outbreaks and shifting clusters, highlight the need for adaptive surveillance systems that integrate spatial analysis to anticipate high-risk areas [49]. However, such strategies may be more costly and challenging to implement [50].

This research aligns with Sustainable Development Goal (SDG) 3: Good Health and Well-Being, which aims to ensure healthy lives and promote well-being for all ages. Specifically, it addresses Target 3.3, which seeks to end epidemics of communicable diseases, including dengue, by 2030. The findings highlight the growing burden of dengue in Mexico, emphasizing the need for coordinated public health responses to reduce transmission, mitigate its impact, and prevent further geographic expansion. This work underscores the importance of strengthening healthcare systems, improving vector control strategies, and enhancing disease surveillance to achieve SDG 3.

The presented findings emphasize the importance of tailoring vector control programs and public health interventions to specific regional contexts, addressing underlying determinants such as urbanization, climate variability, and health infrastructure disparities [51,52].

Effective dengue prevention relies on controlling mosquito populations. Environmental management plays a key role, and communities should be encouraged to eliminate or treat standing water in containers, tires, and other breeding sites through regular clean-up campaigns, especially in urban and peri-urban areas [53]. Urban planning should incorporate dengue prevention by improving drainage systems and reducing water stagnation in construction sites and public spaces [54].

Early warning systems are crucial for predicting and mitigating outbreaks [55]. Climate and environmental monitoring provide valuable insights, while real-time digital surveillance platforms enable healthcare workers and communities to report dengue cases and mosquito breeding sites promptly [56].

Public health campaigns raise awareness and promote preventive behaviors [57]. Integrating dengue education into school curricula can teach children about mosquito biology, breeding sites, and protective measures [58]. Community workshops in high-risk neighborhoods can inform residents about dengue symptoms, prevention, and the importance of early treatment [59].

Regional collaboration also enhances dengue control efforts. Sharing surveillance data, research findings, and best practices can improve response strategies. Joint research initiatives can focus on innovative vector control methods, such as genetically modified mosquitoes or novel insecticides [60].

Sustaining these efforts requires capacity building. Training healthcare workers, entomologists, and public health officials in surveillance, outbreak response, and vector control can strengthen local capabilities [61]. Adequate funding and resources are essential, particularly in low-income regions with high disease burdens.

This study has several limitations that must be acknowledged. First, the analysis relied on reported cases, which may be affected by underreporting or misclassification, particularly in remote areas with limited diagnostic capacity. Second, while the study analyzed spatial and temporal patterns, the data were aggregated at the municipality level, avoiding a more detailed analysis at finer spatial scales, such as the colony or neighborhood level. Third, further research is necessary to investigate the underlying drivers of these patterns, including climatic, socioeconomic, and vector-related factors.

Fourth, approximately three-quarters of the analyzed cases were classified as suspected. According to normative guidelines, this classification applies to cases meeting the clinical criteria for dengue fever but lacking serological or molecular confirmation due to their ambulatory status, absence of warning or severe symptoms, and occurrence during periods of high viral transmission [62]. Although these cases are highly likely to represent true DENV infections [63,64], the absence of laboratory confirmation introduces some uncertainty.

Finally, in our analysis, the kNN parameter was fixed at 5. This choice defines spatial relationships based on the five nearest municipalities, ensuring that each area has a consistent number of neighbors, even in regions with sparse or irregular distributions [65]. While kNN does not account for varying distances between municipalities, resulting in smaller geographic areas being represented in densely populated regions compared to sparsely populated ones, it remains a widely used standard in spatial epidemiology. The selection of kNN = 5 has proven effective in identifying clusters in epidemiological studies, providing a reliable and practical starting point for this analysis [66].

## Conclusions

Our study highlights a growing burden of dengue in Mexico, marked by increasing incidence rates, expanding geographic spread, and shifting spatial clustering patterns over the years. These findings emphasize the urgent need for a coordinated public health response to mitigate the rising impact of dengue and to prevent the further expansion of transmission areas, particularly in regions newly at risk.

## Supporting information

**S1 Table. Dengue incidence rates by municipality, 2022–2024.**
(CSV)

**S2 Table. Clustering data, 2022–2024. Moran's I statistic, Z-scores, and geographic coordinates are presented.**
(XLS)

## Acknowledgments

We would like to thank the Health Research Coordination of the Mexican Social Security Institute for all the support provided in conducting this research.

## Author contributions

**Conceptualization:** Oliver Mendoza-Cano, Efrén Murillo-Zamora.

**Data curation:** Oliver Mendoza-Cano.

**Formal analysis:** Oliver Mendoza-Cano, Rogelio Danis-Lozano, Efrén Murillo-Zamora.

**Investigation:** Yolitzy Cárdenas, Jesús Venegas-Ramírez.

**Methodology:** Rogelio Danis-Lozano, Efrén Murillo-Zamora.

**Resources:** Eder Fernando Ríos-Bracamontes, Luis A. García-Solórzano, Arlette A. Camacho-delaCruz.

**Software:** Oliver Mendoza-Cano.

**Validation:** Xóchitl Trujillo, Miguel Huerta, Mónica Ríos-Silva, Agustin Lugo-Radillo, Jaime Alberto Bricio-Barrios, Verónica Benites-Godínez, Herguin Benjamin Cuevas-Arellano, Juan Manuel Uribe-Ramos, Ramón Solano-Barajas.

**Visualization:** Xóchitl Trujillo.

**Writing – original draft:** Oliver Mendoza-Cano, Efrén Murillo-Zamora.

**Writing – review & editing:** Rogelio Danis-Lozano, Xóchitl Trujillo, Miguel Huerta, Mónica Ríos-Silva, Agustin Lugo-Radillo, Jaime Alberto Bricio-Barrios, Verónica Benites-Godínez, Herguin Benjamin Cuevas-Arellano, Juan Manuel Uribe-Ramos, Ramón Solano-Barajas, Yolitzy Cárdenas, Jesús Venegas-Ramírez, Eder Fernando Ríos-Bracamontes, Luis A. García-Solórzano, Arlette A. Camacho-delaCruz.

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
