## [Decision Letter · Decision Letter 0]

11 Feb 2025

PONE-D-24-59480Spatial patterns and clustering of dengue incidence in Mexico: Analysis of Moran's index across 2,471 municipalities from 2022 to 2024PLOS ONE

Dear Dr. Murillo-Zamora,

Thank you for submitting your manuscript to PLOS ONE. After careful consideration, we feel that it has merit but does not fully meet PLOS ONE’s publication criteria as it currently stands. Therefore, we invite you to submit a revised version of the manuscript that addresses the points raised during the review process.

We look forward to receiving your revised manuscript.

Kind regards,

Harapan Harapan, MD, PhD

Academic Editor

PLOS ONE

Journal Requirements:

2. We note that Figure 1, 3, Supplementary data 1 in your submission contain [map/satellite] images which may be copyrighted. All PLOS content is published under the Creative Commons Attribution License (CC BY 4.0), which means that the manuscript, images, and Supporting Information files will be freely available online, and any third party is permitted to access, download, copy, distribute, and use these materials in any way, even commercially, with proper attribution. For these reasons, we cannot publish previously copyrighted maps or satellite images created using proprietary data, such as Google software (Google Maps, Street View, and Earth). For more information, see our copyright guidelines: http://journals.plos.org/plosone/s/licenses-and-copyright.

a. You may seek permission from the original copyright holder of Figure 1, 3, Supplementary data 1 to publish the content specifically under the CC BY 4.0 license.

3. Please include a caption for figure 2.

4. Please upload a copy of Figure 2 to which you refer in your text on page 7, 9. If the figure is no longer to be included as part of the submission please remove all reference to it within the text.

Reviewers' comments:

Reviewer's Responses to Questions

**Comments to the Author**

1. Is the manuscript technically sound, and do the data support the conclusions?

Reviewer #1: Yes

Reviewer #2: Partly

2. Has the statistical analysis been performed appropriately and rigorously? 

Reviewer #1: Yes

Reviewer #2: Yes

3. Have the authors made all data underlying the findings in their manuscript fully available?

Reviewer #1: Yes

Reviewer #2: Yes

4. Is the manuscript presented in an intelligible fashion and written in standard English?

Reviewer #1: Yes

Reviewer #2: Yes

5. Review Comments to the Author

Reviewer #1: [General Comments]

This manuscript presents an important analysis of dengue transmission dynamics in Mexico, offering valuable epidemiological insights. The spatial and statistical methods applied are appropriate, and the results provide actionable information for disease control. However, certain areas require improvement in clarity, methodological transparency, and discussion depth.

[Specific Comments] – Major

Comment 1: Kindly expand the description of spatial clustering methods. The rationale for using a fixed kNN of 5 should be justified in the context of Mexico’s geographic and epidemiological diversity.

Comment 2: The authors have documented a shift in high-incidence areas from southern and central regions to the southeast and Pacific coast. I suggest you explore more potential drivers such as climate variability, vector control measures, or urbanization patterns.

Comment 3: More discussion is needed regarding how these spatial trends compare with previous dengue outbreaks in Mexico and globally.

Comment 4: The transition from DENV-2 dominance in 2022 to DENV-3 prevalence in 2023-2024 is an important finding. The authors should please discuss possible virological, ecological, or immunological factors influencing this shift.

Comment 5: Building on Comment 4, the authors should consider specific potential public health implications of serotype shifts, including changes in disease severity or vaccine effectiveness.

Comment 6: The discussion on intervention strategies remains general. The authors should kindly provide specific recommendations for vector control, early warning systems, or targeted public health campaigns in the identified high-risk regions.

[Specific Comments] – Minor

Comment 7: Clarify how missing data were handled in dengue incidence reporting.

Comment 8: Please kindly clarify whether population estimates were interpolated for 2023 and 2024 or based on projections.

Comment 9: Report confidence intervals for incidence rates where applicable.

Comment 10: Provide effect sizes alongside p-values in statistical comparisons.

Reviewer #2: A review report of the manuscript entitled “Spatial patterns and clustering of dengue incidence in Mexico: Analysis of Moran's index across 2,471 municipalities from 2022 to 2024”

- (page 9, line 2) in the beginning of the abstract part, what is known or contextual background of this study should be provided.

- (page 9, line 3) the source of the dengue case data is not mentioned. Was the data obtained from health records, a surveillance system, or another source? Please specify if the data is from a national surveillance database or other repositories.

- (page 10, introduction) although the serotype dynamics are addressed in the abstract, the introduction does not mention how the serotypes (e.g., DENV-1 to DENV-4) influence spatial and temporal patterns. Kindly explain the relevance of serotype shifts or dominance to dengue epidemiology.

- (page 10, line 27) the role of climate (e.g., temperature, rainfall) in driving dengue transmission is not mentioned. These are key factors influencing vector activity and virus replication rates and should be briefly noted. Please mention how temperature, rainfall, and vegetation influence vector ecology and dengue transmission. For instance, "Dengue transmission is influenced by climatic factors such as temperature and rainfall, which directly affect mosquito breeding, survival, and virus replication." This can be cited from a study by Dhewantara et al. entitled “Decline of notified dengue infections in Indonesia in 2017: Discussion of the possible determinants.”

- (page 11, line 47) a clear research hypothesis is missing. Please include a statement of your hypothesis in the end of your introduction.

- (page 11, line 58) the method for addressing missing or incomplete case data is not discussed. For instance, were municipalities with incomplete case or population data excluded? Was imputation performed for missing weekly case counts?

- (page 11, line 69) the GIS software or package used for mapping is not mentioned. Was mapping performed in R (e.g., using sf, tmap, or ggplot2), QGIS, or ArcGIS?

- (page 19, line 180) the discussion could benefit from comparisons with other regions where similar spatial or temporal dengue patterns have been observed. For example, how do the patterns in Mexico compare with other endemic regions like Brazil, Southeast Asia, or the Caribbean?

- (page 10, lines 26-28) this statement “This can be attributed to various factors, including human behavior, and environmental conditions (ecological, entomological, infrastructural, and social)” will be stronger if it’s supported by another relevant study. For example, a study by Musa et al. entitled “Revitalizing the state of primary healthcare towards achieving universal health coverage in conflict affected fragile northeastern Nigeria: Challenges, strategies and way forward.”

6. PLOS authors have the option to publish the peer review history of their article (what does this mean? ). If published, this will include your full peer review and any attached files.

**Do you want your identity to be public for this peer review?** For information about this choice, including consent withdrawal, please see our Privacy Policy .

Reviewer #1: No

Reviewer #2: No

---

## [Author Response · Author response to Decision Letter 0]

18 Mar 2025

REVIEWER 1

General Comment

COMMENT 1: This manuscript presents an important analysis of dengue transmission dynamics in Mexico, offering valuable epidemiological insights. The spatial and statistical methods applied are appropriate, and the results provide actionable information for disease control. However, certain areas require improvement in clarity, methodological transparency, and discussion depth.

RESPONSE (R): We sincerely thank Reviewer 1 for the insightful comments and positive feedback on our manuscript. We appreciate your recognition of the significance of our analysis and its contribution to understanding the dynamics of dengue transmission in Mexico. In response to your suggestions, we have addressed the areas requiring improvement.

[Specific Comments] – Major

COMMENT 2: Kindly expand the description of spatial clustering methods. The rationale for using a fixed kNN of 5 should be justified in the context of Mexico’s geographic and epidemiological diversity.

R: We agree with the suggested specification. This choice ensured that each region is compared with a consistent number of neighboring areas, despite variations in population density and transmission patterns, thereby enhancing the reliability of identifying spatial clusters across diverse settings. This is now included in the new version of the manuscript submitted for revision (please refer to lines 95-99).

COMMENT 3: The authors have documented a shift in high-incidence areas from southern and central regions to the southeast and Pacific coast. I suggest you explore more potential drivers such as climate variability, vector control measures, or urbanization patterns.

R: In agreement with Reviewer 1's suggestion, we have expanded the discussion on potential drivers in the revised manuscript (please refer to lines 233-239).

COMMENT 4: More discussion is needed regarding how these spatial trends compare with previous dengue outbreaks in Mexico and globally.

R: In the new version of the manuscript submitted for revision, a broader discussion of the spatial trends of dengue in Mexico and other countries and regions is included (please refer to lines 212-232).

COMMENT 5: The transition from DENV-2 dominance in 2022 to DENV-3 prevalence in 2023-2024 is an important finding. The authors should please discuss possible virological, ecological, or immunological factors influencing this shift.

R: This shift may be linked to the reemergence of DENV-3, as its circulation had not been documented in the country for over a decade. This, along with other potential drivers, is discussed in the revised manuscript (please refer to lines 242-247).

COMMENT 6: Building on Comment 5, the authors should consider specific potential public health implications of serotype shifts, including changes in disease severity or vaccine effectiveness.

R: As suggested, we addressed how these shifts can influence disease severity, particularly in populations with prior heterotypic immunity, and impact vaccine effectiveness. This is now included in the new version of the manuscript submitted for revision (please refer to lines 248-254).

COMMENT 6: The discussion on intervention strategies remains general. The authors should kindly provide specific recommendations for vector control, early warning systems, or targeted public health campaigns in the identified high-risk regions.

R: We agree with Reviewer 1 and have incorporated a broader discussion on intervention strategies in the revised manuscript submitted for revision (please refer to lines 273–295).

[Specific Comments] – Minor

COMMENT 7: Clarify how missing data were handled in dengue incidence reporting.

R: Municipalities with missing incidence data were excluded. To facilitate spatial comparisons, municipalities were categorized into quintiles based on the computed rates. This is now included in the new version of the manuscript submitted for revision (please refer to lines 65-67).

COMMENT 8: Please kindly clarify whether population estimates were interpolated for 2023 and 2024 or based on projections.

R: Population data were obtained from publicly available projections (2021-2040) by the National Population Council of Mexico. This is now included in the new version of the manuscript submitted for revision (please refer to lines 75-78).

COMMENT 9: Report confidence intervals for incidence rates where applicable.

R: We appreciate your suggestion regarding the inclusion of confidence intervals (CIs) for the municipality-specific dengue rates. In this study, the rates were calculated using population-level epidemiological surveillance data, meaning they represent complete values rather than sample-based estimates subject to statistical variability.

COMMENT 10: Provide effect sizes alongside p-values in statistical comparisons.

R: We agree with Reviewer 1. The magnitude of Moran’s I represents the extent of spatial autocorrelation, with values approaching +1 indicating strong clustering and those nearing -1 signifying pronounced dispersion. This is now specified in the new version of the manuscript submitted for revision (please refer to lines 149-151).

REVIEWER 2

General Comment

COMMENT 1: A review report of the manuscript entitled “Spatial patterns and clustering of dengue incidence in Mexico: Analysis of Moran's index across 2,471 municipalities from 2022 to 2024”.

Response (R): We sincerely appreciate the thoughtful observations and valuable suggestions provided in the review report. We have carefully considered all comments and implemented the necessary revisions to improve the clarity, rigor, and overall quality of the manuscript.

Specific Comments

COMMENT 2: (page 9, line 2) in the beginning of the abstract part, what is known, or contextual background of this study should be provided.

R: As suggested by Reviewer 2, a brief contextual background has been included in the new version of the manuscript submitted for revision (please refer to lines 30-34).

COMMENT 3: (page 9, line 3) the source of the dengue case data is not mentioned. Was the data obtained from health records, a surveillance system, or another source? Please specify if the data is from a national surveillance database or other repositories.

R: We apologize for the previous omission. The data were obtained through the normative epidemiological surveillance system for vector-borne diseases. This is now specified in the new version of the manuscript submitted for revision (please refer to lines 66-67).

COMMENT 4: (page 10, introduction) although the serotype dynamics are addressed in the abstract, the introduction does not mention how the serotypes (e.g., DENV-1 to DENV-4) influence spatial and temporal patterns. Kindly explain the relevance of serotype shifts or dominance to dengue epidemiology.

R: We agree with Reviewer 2, and the revised Introduction now includes a brief description of the spatial and temporal patterns of DENV serotypes, along with the relevance of serotype shifts to dengue epidemiology (please refer to lines 41-43).

COMMENT 5: (page 10, line 27) the role of climate (e.g., temperature, rainfall) in driving dengue transmission is not mentioned. These are key factors influencing vector activity and virus replication rates and should be briefly noted. Please mention how temperature, rainfall, and vegetation influence vector ecology and dengue transmission. For instance, "Dengue transmission is influenced by climatic factors such as temperature and rainfall, which directly affect mosquito breeding, survival, and virus replication." This can be cited from a study by Dhewantara et al. entitled “Decline of notified dengue infections in Indonesia in 2017: Discussion of the possible determinants.”

R: We thank Reviewer 2 for the suggested reference, which the research group found pertinent and has now been included in the revised manuscript. Additionally, a brief description of the role of climatic factors in dengue epidemiology has been added (please refer to lines 30-34).

COMMENT 6: (page 11, line 47) a clear research hypothesis is missing. Please include a statement of your hypothesis in the end of your introduction.

R: We hypothesized that dengue transmission hotspots are not randomly distributed but follow spatial patterns influenced by disease dynamics over time. Specifically, we expect that municipalities with historically high dengue incidence will exhibit persistent transmission, with spatial clustering reflecting underlying transmission trends. This is now included in the new version of the manuscript submitted for revision (please refer to lines 58-61). The hypothesis is discussed in lines 197-200 and 209-211.

COMMENT 7: (page 11, line 58) the method for addressing missing or incomplete case data is not discussed. For instance, were municipalities with incomplete case or population data excluded? Was imputation performed for missing weekly case counts?

R: No imputations were performed, and municipalities with missing incidence data were excluded. This is now stated in the new version of the manuscript submitted for revision (please refer to lines 76-78).

COMMENT 8: (page 11, line 69) the GIS software or package used for mapping is not mentioned. Was mapping performed in R (e.g., using sf, tmap, or ggplot2), QGIS, or ArcGIS?

R: Mapping was performed in R 4.4.1 using the ggplot2 package. This is now stated in the new version of the manuscript submitted for revision (please refer to line 80-84).

COMMENT 9: (page 19, line 180) the discussion could benefit from comparisons with other regions where similar spatial or temporal dengue patterns have been observed. For example, how do the patterns in Mexico compare with other endemic regions like Brazil, Southeast Asia, or the Caribbean?

R: A comparison of the observed patterns with those in other regions has been included in the revised version of the manuscript submitted for review (please refer to lines 212–239).

COMMENT 10: (page 10, lines 26-28) this statement “This can be attributed to various factors, including human behavior, and environmental conditions (ecological, entomological, infrastructural, and social)” will be stronger if it’s supported by another relevant study. For example, a study by Musa et al. entitled “Revitalizing the state of primary healthcare towards achieving universal health coverage in conflict affected fragile northeastern Nigeria: Challenges, strategies and way forward.”

R: We agree with Reviewer 2 on the importance of supporting the statement. After carefully analyzing the suggested reference, we have included it in the revised version of the manuscript submitted for review (please refer to line 31).

1. Is the manuscript technically sound, and do the data support the conclusions?

Reviewer #1: Yes

Reviewer #2: Partly

2. Has the statistical analysis been performed appropriately and rigorously?

Reviewer #1: Yes

Reviewer #2: Yes

3. Have the authors made all data underlying the findings in their manuscript fully available?

Reviewer #1: Yes

Reviewer #2: Yes

4. Is the manuscript presented in an intelligible fashion and written in standard English?

Reviewer #1: Yes

Reviewer #2: Yes

6. PLOS authors have the option to publish the peer review history of their article (what does this mean?). If published, this will include your full peer review and any attached files.

Do you want your identity to be public for this peer review? For information about this choice, including consent withdrawal, please see our Privacy Policy.

Reviewer #1: No

Reviewer #2: No

Journal Requirements

COMMENT 1: Please ensure that your manuscript meets PLOS ONE's style requirements, including those for file naming. The PLOS ONE style templates can be found at https://journals.plos.org/plosone/s/file?id=wjVg/PLOSOne_formatting_sample_main_body.pdf and https://journals.plos.org/plosone/s/file?id=ba62/PLOSOne_formatting_sample_title_authors_affiliations.pdf

Response (R): I appreciate the observation, and the relevant modifications were made according to the guidelines.

COMMENT 2: We note that Figure 1, 3, Supplementary data 1 in your submission contain [map/satellite] images which may be copyrighted. All PLOS content is published under the Creative Commons Attribution License (CC BY 4.0), which means that the manuscript, images, and Supporting Information files will be freely available online, and any third party is permitted to access, download, copy, distribute, and use these materials in any way, even commercially, with proper attribution. For these reasons, we cannot publish previously copyrighted maps or satellite images created using proprietary data, such as Google software (Google Maps, Street View, and Earth). For more information, see our copyright guidelines: http://journals.plos.org/plosone/s/licenses-and-copyright.

a. You may seek permission from the original copyright holder of Figure 1, 3, Supplementary data 1 to publish the content specifically under the CC BY 4.0 license.

We recommend that you contact the original copyright holder with the Content Permission Form (http://journals.plos.org/plosone/s/file?id=7c09/content-permission-form.pdf) and the following text: “I request permission for the open-access journal PLOS ONE to publish XXX under the Creative Commons Attribution License (CCAL) CC BY 4.0 (http://creativecommons.org/licenses/by/4.0/). Please be aware that this license allows unrestricted use and distribution, even commercially, by third parties. Please reply and provide explicit written permission to publish XXX under a CC BY license and complete the attached form.”

USGS National Map Viewer (public domain): http://viewer.nationalma

---

## [Decision Letter · Decision Letter 1]

2 May 2025

Spatial patterns and clustering of dengue incidence in Mexico: Analysis of Moran's index across 2,471 municipalities from 2022 to 2024

PONE-D-24-59480R1

Dear Dr. Murillo-Zamora,

We’re pleased to inform you that your manuscript has been judged scientifically suitable for publication and will be formally accepted for publication once it meets all outstanding technical requirements.

Kind regards,

Harapan Harapan, MD, PhD

Academic Editor

PLOS ONE

Additional Editor Comments (optional):

Reviewers' comments:

Reviewer's Responses to Questions

**Comments to the Author**

1. If the authors have adequately addressed your comments raised in a previous round of review and you feel that this manuscript is now acceptable for publication, you may indicate that here to bypass the “Comments to the Author” section, enter your conflict of interest statement in the “Confidential to Editor” section, and submit your "Accept" recommendation.

Reviewer #1: All comments have been addressed

Reviewer #2: All comments have been addressed

2. Is the manuscript technically sound, and do the data support the conclusions?

Reviewer #1: Yes

Reviewer #2: Yes

3. Has the statistical analysis been performed appropriately and rigorously? 

Reviewer #1: Yes

Reviewer #2: Yes

4. Have the authors made all data underlying the findings in their manuscript fully available?

Reviewer #1: (No Response)

Reviewer #2: Yes

5. Is the manuscript presented in an intelligible fashion and written in standard English?

Reviewer #1: Yes

Reviewer #2: Yes

6. Review Comments to the Author

Reviewer #1: (No Response)

Reviewer #2: The objective of this study was to analyze spatial and temporal patterns of dengue incidence across municipalities in Mexico from 2022 to 2024. Though I do hope further studies could explore the spatial regression, but the overall results presented are sufficient to answer the research question. I have no further concerns over the manuscript.

7. PLOS authors have the option to publish the peer review history of their article (what does this mean? ). If published, this will include your full peer review and any attached files.

**Do you want your identity to be public for this peer review?** For information about this choice, including consent withdrawal, please see our Privacy Policy .

Reviewer #1: **Yes: ** Oluwafemi Balogun, MBBS, MPH

Reviewer #2: No

---

## [Editor Report · Acceptance letter]

PONE-D-24-59480R1

PLOS ONE

Dear Dr. Murillo-Zamora,

I'm pleased to inform you that your manuscript has been deemed suitable for publication in PLOS ONE. Congratulations! Your manuscript is now being handed over to our production team.

Kind regards,

on behalf of

Dr. Harapan Harapan

Academic Editor

PLOS ONE